# SaFiRe: Saccade-Fixation Reiteration with Mamba for Referring Image Segmentation

**Zhenjie Mao**[1]    **Yuhuan Yang**[1]    **Chaofan Ma**[1]    **Dongsheng Jiang**[4]
**Jiangchao Yao**[1†]    **Ya Zhang**[2,3†]    **Yanfeng Wang**[2]

[1]Cooperative Medianet Innovation Center, Shanghai Jiao Tong University
[2]School of Artificial Intelligence, Shanghai Jiao Tong University
[3]Institute of Artificial Intelligence for Medicine, Shanghai Jiao Tong University School of Medicine
[4]Huawei Inc.
{zhenjmao, yangyuhuan, chaofanma, sunarker, ya_zhang, wangyanfeng622}@sjtu.edu.cn
dongsheng_jiang@outlook.com

## Abstract

Referring Image Segmentation (RIS) aims to segment the target object in an image given a natural language expression. While recent methods leverage pre-trained vision backbones and more training corpus to achieve impressive results, they predominantly focus on simple expressions—short, clear noun phrases like "red car" or "left girl". This simplification often reduces RIS to a key word/concept matching problem, limiting the model's ability to handle referential ambiguity in expressions. In this work, we identify two challenging real-world scenarios: *object-distracting expressions*, which involve multiple entities with contextual cues, and *category-implicit expressions*, where the object class is not explicitly stated. To address the challenges, we propose a novel framework, **SaFiRe**, which mimics the human two-phase cognitive process—first forming a global understanding, then refining it through detail-oriented inspection. This is naturally supported by Mamba's scan-then-update property, which aligns with our phased design and enables efficient multi-cycle refinement with linear complexity. We further introduce **aRefCOCO**, a new benchmark designed to evaluate RIS models under ambiguous referring expressions. Extensive experiments on both standard and proposed datasets demonstrate the superiority of SaFiRe over state-of-the-art baselines. Project page: https://zhenjiemao.github.io/SaFiRe/.

## 1   Introduction

Referring Image Segmentation (RIS), aiming to segment the target object in an image based on the textual description, is a fundamental task in multi-modal understanding. Most studies solve the RIS task by designing image-text alignment modules to bridge the modality gap [1–3]. With the success of image foundation models [4–6], recent works further improved the performance of RIS by leveraging the pre-trained image features [7–9] and introducing more training corpus [10–12].

Despite significant process, current RIS methods primarily focus on a *narrow* scenario, where most of the referring expressions consist of *a single, simple noun phrase* that *clearly* identifies the target object, *e.g.*, "left girl" [13], "red car" [13], etc. We summarize this as the **simple expression** pattern. Under this paradigm, the RIS task is often over-simplified and modeled as a "key word/concept matching problem". The model only needs to extract the key words or concepts from the given text, then matches them with the image patches. Most existing RIS methods adopt this simplified

---

[†]Corresponding authors.

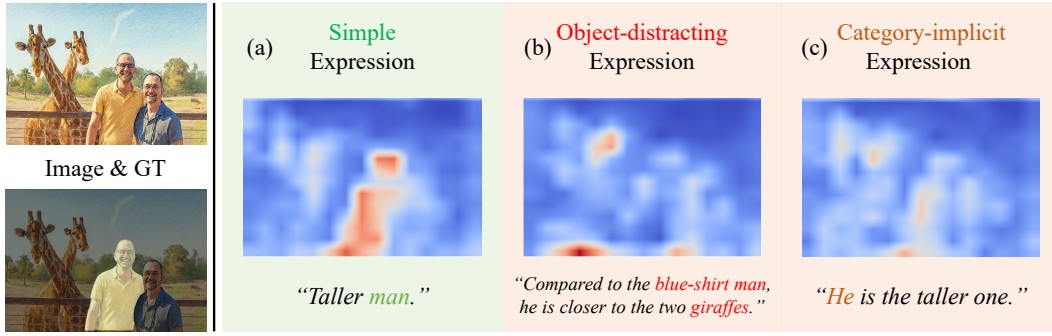

Figure 1: **Referential Ambiguity: One Object, Divergent Attention.** Attention maps under three types of referring expressions, all targeting the same object. **(a) Simple expression** yields accurate focus. **(b) Object-distracting expression** misguides attention to irrelevant regions. **(c) Category-implicit expression** leads to dispersed attention. This highlights the challenge of referential ambiguity for "key word/concept matching" method and motivates our saccade-fixation framework.

formulation. For example, LAVT [7] and ReLA [14] treat RIS as a process of matching individual words to specific pixels, matching keywords for each image location and segmenting the regions most relevant. VLT [15] introduces multiple queries to focus on various word pairs, and it reduces the rich semantics of language to a set of discrete choices. Similarly, MagNet [16] tries to emphasize key words by masking several input textual tokens and matching them with visual objects.

Although these methods demonstrate reasonable performance to some extent in this simple expression scenario, we notice that, in real-world applications, referring expressions often exhibit **referential ambiguity**, where the mapping between the text and the target object is indirect, underspecified, or contextually entangled. We summarize this into two challenging cases. (1) One is **object-distracting expression**, where the referring sentence contains multiple noun phrases but only one is the true referent, *e.g.*, "compared to the blue-shirt man, he is closer to the two giraffes". This introduces misleading contextual cues that can divert attention away from the actual target, increasing the difficulty of accurate segmentation. (2) The other is **category-implicit expression**, where the target object lacks an explicit categorical attribute, *e.g.*, "he is the taller one". This is common in spoken language, where pronouns and comparative or other distinct adjectives are frequently used without specifying object categories, making it difficult to determine the referent based solely on class-level semantics. As shown in Fig. 1, we visualize the attention maps under three types of expressions that refer to the same object. While the model successfully highlights the target under the simple expression (a), its focus becomes diffuse and uncertain under the object-distracting (b) and category-implicit (c) expressions. The referring expression in these cases is not ideally with a single object or a clear category, resulting in limited comprehension and weak visual-textual interaction for the existing methods that rely on key word/concept matching.

To address this issue, we investigate how humans interpret the correspondence between complex textual descriptions and visual representations, and how they identify objects within images. According to the research in cognitive psychology [17, 18], humans engage in a two-phase process. The first phase, which can be summarized as *global semantic understanding*, involves an initial overview of the sentence and a general scan of the image. The second phase, referred to as *cross-modal refinement*, occurs when the individual focuses on finer details, re-examining the sentence and systematically inspecting each region of the image to extract relevant information. This dual-phase process underscores how humans effectively integrate textual and visual information.

Inspired by this, we propose our network **SaFiRe** with two *alternating* operations: **Saccade** and **Fixation**, simulating the two human cognition phase respectively. This formulation aligns well with state-space sequence modeling, and we adopt Mamba as the underlying architecture to support our framework in a cognitively inspired and structurally efficient manner. Specifically, the **Saccade** operation performs a rapid "glimpse" of both the text and image. By modulating the image features with the general meaning of the text, this operation establishes a rough correspondence between the modalities after SSM scanning. While the **Fixation** operation aims to refine this coarse result by carefully examining the whole image region by region, and continuously reconfirms the text during this process. To this end, this operation alternates each image region with reiterated text in

a unified sequence for SSM scanning, enabling Mamba to focus locally while maintaining global context—naturally leveraging its order-sensitive and memory-efficient design. Furthermore, with the *reiteration* of the two-phase process, the model is able to progressively refine its cross-modal understanding and accurately identify the target referent.

To sum up, our contributions lie in four folds:

• We highlight the limitation of recent RIS methods, identifying their tendency to oversimplify RIS as a key word/concept matching problem, which often fails when confronted with referential ambiguity.

• We propose **SaFiRe**, a novel RIS framework inspired by cognitive psychology, which simulates the human cognitive process through alternating two inspection phases to achieve coarse-to-fine alignment and comprehensive cross-modal understanding.

• We design two operations—**Saccade** and **Fixation**—where Saccade establishes initial semantic correspondence through fast global scanning, and Fixation enables fine-grained region-wise inspection guided by reiterated textual cues for more accurate segmentation.

• To evaluate the effectiveness of our approach, we not only conduct experiments on the traditional RefCOCO family of datasets but also introduce a new test-only benchmark, **aRefCOCO**, which features ambiguous expressions. Extensive experiments and ablation studies demonstrate the superior performance of our method over the state-of-the-art methods.

## 2  Related Work

**Referring Image Segmentation** (RIS) is a challenging task that requires identifying and segmenting specific objects in images based on natural language expressions. Early methods mainly focus on attention-based vision-language fusion strategies. For example, VLT [15] introduced a cross-attention-based framework with query generation, while LAVT [7] demonstrated that encoder-level cross-modal fusion significantly enhances vision-language alignment. ReLA [14] advanced the field by incorporating fine-grained cross-region modality interactions. Later methods such as CGFormer [11] and MagNet [16] enhance segmentation accuracy by integrating object-level priors and mask supervision. The advent of Large Language Models (LLMs) has also spurred the development of LLM-based RIS models [19–21], leveraging their powerful multi-modal reasoning abilities to better address the RIS challenges. While existing methods perform well on simple expressions with clear object references, they often treat RIS as a key word/concept matching problem, which limits their ability to handle referential ambiguity.

**State Space Models and Multi-Modal Extensions.** Originally developed for dynamical systems, state-space models (SSMs) have proven effective in deep learning for modeling long-range dependencies. S4 [22] demonstrated their scalability, and Mamba [23] further improved efficiency and performance through a novel selection mechanism. Mamba has achieved strong results across vision [6, 24, 25] and language [26] tasks. Recently, its use in multi-modal learning [27–31] has gained interest, mainly in structurally aligned settings. However, current approaches offer limited cross-modal alignment. For instance, Coupled Mamba [32] uses shallow fusion strategies, while ReMamber [9] focuses on encoder-level fusion but oversimplifies local interaction as a keyword-matching problem. To address these gaps, we propose a framework that leverages Mamba's sequential processing to emulate the human-like two-phase grounding process.

## 3  Method

### 3.1  Overview

The standard definition of RIS task is to segment the target object based on the input textual description. Given an image $\mathbf{I}$ and a textual description $\mathbf{T}$, we aim to predict a binary mask $\mathbf{M}$ that indicates the referring target in the image. The overall framework of our proposed can be written as:

$$\Phi_{\textbf{SaFiRe}}(\mathbf{I}, \mathbf{T}) = \mathbf{M}. \tag{1}$$

For encoders, we use Swin-Transformer [4] and BERT [33] to extract image and text features respectively,

$$\mathbf{F}_v^0 = \mathcal{E}_v(\mathbf{I}) \in \mathbb{R}^{H \times W \times C}, \quad \mathbf{F}_t^0 = \mathcal{E}_t(\mathbf{T}) \in \mathbb{R}^{L \times C}. \tag{2}$$

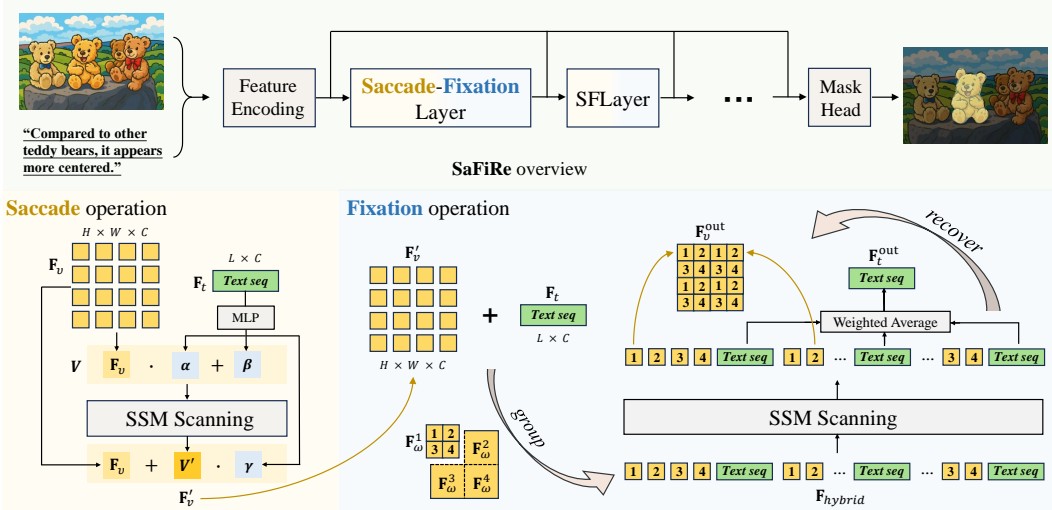

Figure 2: **Overview of the SaFiRe Architecture.** For each SFLayer, it consists of Saccade operation and Fixation operation. The **Saccade** operation corresponds to the phase of global semantic understanding. It enables the model to rapidly scan both visual and textual inputs, establishing a coarse-level alignment between the two modalities. The **Fixation** operation mirrors the cross-modal refinement phase. It allows the model to attend to specific local visual regions while re-examining the textual input, facilitating the extraction of fine-grained, task-relevant information.

Then, we decode the image and text features through a series of our carefully designed **SFLayers**:

$$\mathbf{F}_v^{i+1}, \mathbf{F}_t^{i+1} = \Phi_{\text{SFLayer}}^{(i)}(\mathbf{F}_v^i, \mathbf{F}_t^i), \quad i \in \{0, 1, 2, 3\}. \tag{3}$$

Finally we predict mask with processed image features:

$$\mathbf{M} = \Phi_{\text{mask}}(\mathbf{F}_v^1, \mathbf{F}_v^2, \mathbf{F}_v^3, \mathbf{F}_v^4). \tag{4}$$

**SFLayer Design.** Our **SaFiRe** framework simulates the two-phase human cognitive process by applying a specialized module called the *Saccade-Fixation Layer (SFLayer)*. Each SFLayer consists of two sequential operations: **Saccade** and **Fixation**. The **Saccade** operation corresponds to the phase of global semantic understanding. It enables the model to rapidly scan both visual and textual inputs, establishing a coarse-level alignment between the two modalities. In contrast, the **Fixation** operation mirrors the cross-modal refinement phase. It allows the model to attend to specific local visual regions while re-examining the textual input, facilitating the extraction of fine-grained, task-relevant information. Formally, Eq. (3) can be expanded in detail as:

$$\begin{aligned} \mathbf{F}_v^{i+1}, \mathbf{F}_t^{i+1} &= \Phi_{\text{SFLayer}}^{(i)}(\mathbf{F}_v^i, \mathbf{F}_t^i) \\ &= \Phi_{\text{Fixation}}\left(\Phi_{\text{Saccade}}(\mathbf{F}_v^i, \mathbf{F}_t^i)\right). \end{aligned} \tag{5}$$

By *reiteratively* alternating between these two operations across multiple layers, the model progressively enhances its multi-modal understanding. This saccade-fixation reiteration across layers serves as the core mechanism for accurate target localization. For a clearer view of this process, the architectural details of our framework are shown in Fig. 2, and we detail the design of **Saccade** and **Fixation** operations in Sec. 3.2 and Sec. 3.3 respectively.

## 3.2 Saccade: Quick Glimpse of Image-Text Correspondence

Our **Saccade** operation aims to take a *quick glimpse* of both text and image information, and establish a rough correspondence between them. This step is designed to mimic the initial phase of human referential understanding—namely, *a coarse and global semantic alignment* between modalities before any fine-grained reasoning occurs. We implement this by modulating image features with global textual semantics via a lightweight yet effective operation. Concretely, we adopt a DiT-inspired [34] Norm Adaptation, the pooled textual representation adjusts the statistics of image features, in contrast to class-based conditioning in the original design.

To be more specific, the Norm Adaptation operation pools a global vector from textual sequence and adapts the image's scale and bias *globally* according to this vector. In our practice, with image feature $\mathbf{F}_v \in \mathbb{R}^{H \times W \times C}$ and text feature $\mathbf{F}_t \in \mathbb{R}^{L \times C}$, we apply average pooling and linear projection on text side to generate global scale, bias and shift factors:

$$[\alpha, \beta, \gamma] = \text{Linear}(\text{AvgPool}(\mathbf{F}_t)) \in \mathbb{R}^{3 \times C}. \tag{6}$$

The image feature is then adapted with the three factors:

$$\mathbf{F}'_v = \gamma \cdot \text{VSSM}(\alpha \cdot \mathbf{F}_v + \beta) + \mathbf{F}_v \in \mathbb{R}^{H \times W \times C}, \tag{7}$$

where $\text{VSSM}(\cdot)$ denotes the visual SSM feed-forward layer in VMamba [6].

This global adapting operation performs a quick and general interaction between the two modalities, making the image features more prepared for the subsequent **Fixation** operation.

### 3.3 Fixation: Detailed Examination of Cross-Modal Alignment

For **Fixation** operation, we aim to foster image-text communication in a more delicate manner, by checking the image region by region according to the text description to determine the target more accurately. We find that *the SSM mechanism is especially suitable for this*. The hidden states of SSM is updated by previous input and its scanning mechanism is a vivid simulation of human behavior: it focuses more on the recent input sequence while keeping long-term global memory in mind.

Thus, we utilize the most recent VSSM architecture VMamba [6], and came up with a novel **Fixation** operation on top of it. The Fixation operation follows a "**group-scan-recover**" pattern, where it first **(1) groups** the image sequence into non-overlapping local regions via window splitting, and then **(2) scans** each region and the text alternately with SSM, effectively modeling fine-grained cross-modal interactions within a unified multi-modal sequence, finally **(3) recovers** the processed unified multi-modal sequence back into separate image and text representations for the next operation. We here detail each of them in order.

**Group.** We aim to divide image sequence into several regions while maintaining their local structural priors. We draw inspiration from Swin [4] and use similar split mechanism with several non-overlapping windows. To be specific, with input image sequence $\mathbf{F}_v \in \mathbb{R}^{H \times W \times C}$ and window size $w$, we split the sequence into $P = HW/w^2$ windows:

$$\mathbf{F}_v = [\mathbf{F}_w^1, \mathbf{F}_w^2, \cdots, \mathbf{F}_w^P], \quad \mathbf{F}_w^i \in \mathbb{R}^{w^2 \times C}. \tag{8}$$

As SSM itself offers global reception field, we do not apply sliding window mechanism as in Swin.

**Scan.** We aim to perform local and fine-grained scanning while foster cross-modal interaction. There is one key property under SSM mechanism: more recent tokens will be more relevant to the current hidden state than the previous ones. In other words, for SSM *the input order matters*. We make full utilization of this property and reiterate the text sequence multiple times within each image region.

To be specific, with input image feature $\mathbf{F}_v \in \mathbb{R}^{H \times W \times C}$ and text feature $\mathbf{F}_t \in \mathbb{R}^{L \times C}$, we re-arrange a hybrid multi-modal sequence $\mathbf{F}_{hybrid}$ in the following manner:

$$\mathbf{F}_{hybrid} = [\mathbf{F}_w^1, \mathbf{F}_t, \mathbf{F}_w^2, \mathbf{F}_t, \cdots, \mathbf{F}_w^P, \mathbf{F}_t] \in \mathbb{R}^{(HW+PL) \times C}. \tag{9}$$

Then, we use an VSSM layer to process this hybrid sequence:

$$\mathbf{F}'_{hybrid} = \text{VSSM}(\mathbf{F}_{hybrid}) = [\mathbf{F}_w^{1'}, \mathbf{F}_t^{1'}, \mathbf{F}_w^{2'}, \mathbf{F}_t^{2'}, \cdots, \mathbf{F}_w^{P'}, \mathbf{F}_t^{P'}]. \tag{10}$$

Due to the re-arrangement of hybrid sequence and the built-in SSM mechanism, the model will always be focused on the query target with the help of periodically reiterated text tokens. This encourages the model to focus on the description as a coherent and complete whole, preserving its continuity and full meaning, rather than discretizing it into isolated key words/concepts for global matching.

**Recover.** This separates the unified multi-modal sequence back into distinct image and text features, preparing them for the next layer. For image features we re-arrange them, and for text features, we apply weighted average across all repeated sentences as updated representation. The output feature can be expressed as:

$$\mathbf{F}_v^{\text{out}} = [\mathbf{F}_w^{1'}, \mathbf{F}_w^{2'}, \cdots, \mathbf{F}_w^{P'}] \in \mathbb{R}^{H \times W \times C}, \quad \mathbf{F}_t^{\text{out}} = \sum_{o=1}^{P} w_o \mathbf{F}_t^{o'}/P \in \mathbb{R}^{L \times C}. \tag{11}$$

Here $\mathbf{w} = [w_1, \cdots, w_P] \in \mathbb{R}^P$ is a learnable average weight. $\mathbf{F}_v^{\text{out}}$ and $\mathbf{F}_t^{\text{out}}$ are the output features and then passed to the next SFLayer for further processing.

The **Fixation** operation achieves the goal of delicate multi-modal understanding by fully utilizing the underlying mechanism of SSM. By repeating the text multiple times within the image sequence, the hidden state space is periodically refreshed, forcing the model to ***continuously memorize and reinforce*** its segmentation target. With the help of SSM mechanism, model is able to be more focused on ***the current area*** while ensures the ***global receptive field*** for segmentation.

**Discussion: Why Mamba Fits Our Framework.** Our framework focuses on simulating the sequential nature of human cognition process—alternating between global understanding and localized refinement. This requires a model that can processes information step-by-step. Mamba's state-space scanning aligns perfectly with this paradigm, as it processes sequences in a directional, recurrent manner. Besides, its linear complexity also allows efficient, dense region-text interactions, supporting fine-grained reasoning without the overhead of attention. These advantages make Mamba a structurally appropriate and computationally efficient fit for our group-scan-recover Fixation operation and the overall Saccade-Fixation Reiteration framework.

### 3.4 Segmentation Head

For final mask prediction, we combine the multi-level visual output $[\mathbf{F}_v^1, \mathbf{F}_v^2, \mathbf{F}_v^3, \mathbf{F}_v^4]$ of each SFLayer in a top-down manner, following LAVT [7], as characterized by Eq.(4). The training process employs a combination of Dice loss [35] and Focal loss [36] as the composite loss function, as follows:

$$\mathcal{L} = \alpha \cdot \mathcal{L}_{\text{Dice}} + (1 - \alpha) \cdot \mathcal{L}_{\text{Focal}}, \tag{12}$$

where the weighting coefficient $\alpha$ is set to 0.5 by default.

### 3.5 Complexity Analysis

Mamba-based architectures are known for their linear complexity in long sequences processing. However, since we introduce multiple times of text repeating in **Fixation** module, the overall sequence length will be slightly increased. Here we take a concise analysis of the computational complexity of our **Fixation** module. Formally, with image sequence length $HW$, text sequence length $L$ and local window size $w$, the complexity is defined by the overall sequence length:

$$\text{Complexity} = O(HW + HW/w^2 \cdot L) = (1 + L/w^2) \cdot O(HW). \tag{13}$$

In practice, we set window size $w$ to 4, and average text length $L$ to around 15. The overall complexity is thus increased by a constant factor of 0.9, which is nearly negligible.

## 4 Experiments

### 4.1 Datasets and Metrics

**Traditional Benchmarks.** Following prior works [7, 16, 9], we systematically assess model performance on the widely used RefCOCO benchmarks: RefCOCO [13], RefCOCO+ [13] and RefCOCOg [37]. These datasets are all grounded in the MSCOCO [38] visual corpus but differ significantly in their linguistic properties. RefCOCO and RefCOCO+ feature concise referring expressions, averaging 3.6 words per phrase. Notably, RefCOCO+ omits absolute spatial terms (e.g., "left/right" or ordinal indicators), thereby introducing additional difficulty. In contrast, RefCOCOg features more descriptive and elaborate language, with expressions averaging 8.4 words and incorporating more references to locations and appearances, making it the most linguistically complex and challenging benchmark among the three.

**aRefCOCO.** We introduce aRefCOCO (ambiguous RefCOCO), a **test-only benchmark** constructed by reannotating a subset of images from the RefCOCO [13] and RefCOCOg [37] test splits with more challenging referring expressions. It is specifically designed to evaluate model performance on **object-distracting** and **category-implicit** scenarios—two aspects that remain difficult for existing models. The referring expressions in aRefCOCO are significantly more complex, averaging 12.6 words per sentence. To increase the difficulty of category-implicit references, the benchmark includes a higher frequency of pronouns (e.g., "it", "he", "she"), which obscure explicit category cues and demand deeper contextual understanding. To better support object-distracting evaluation, expressions exhibit more frequent use of prepositions (e.g., "and", "to", "of"), introducing richer relational structures

and more noun phrases that increase the likelihood of semantic confusion. Visually, aRefCOCO images contain 3.1 same-category distractors per image on average—87.5% more than RefCOCOg's 1.6—requiring models to resolve denser visual ambiguities for accurate grounding. In terms of scale, aRefCOCO introduces 7,050 text-image pairs generated by *Qwen-VL-Max* [39] and meticulously filtered—a 40% expansion over RefCOCOg's test split (5,023 pairs)—establishing a benchmark that challenges existing RIS models. For more details of aRefCOCO, see Appendix D.

**Metrics.** We adopt oIoU as the main metric and further incorporate mIoU and Precision@X (X$\in \{50, 70, 90\}$) for a more comprehensive evaluation, where Precision@X means the percentage of test images with an IoU score higher than X%.

## 4.2 Implementation Details

Our model employs the Swin Transformer-B [4] and BERT-B [33] as the vision and language encoders, respectively, following prior work [7, 11]. Swin-B is initialized with ImageNet pre-trained weights, and BERT-B uses the official checkpoints. For SFlayers, we adopt the official implementation of the VSSM from VMamba [6] and set the window size to $4 \times 4$ in the Fixation operation. The model is trained end-to-end for 50 epochs using the AdamW optimizer with a learning rate of $5e$-5 and weight decay of $1e$-4, along with a cosine learning rate scheduler. All experiments are conducted on 2 NVIDIA A100 GPUs.

## 4.3 Comparison with the State-of-the-Arts

Table 1: **Comparison of RIS Methods Trained on Single Datasets across RefCOCO, RefCOCO+, and RefCOCOg with oIoU**. U indicates UMD partition of RefCOCOg. The best performances are in **bold**.

| Methods | Encoders | | RefCOCOg (hard) | | RefCOCO+ (medium) | | | RefCOCO (easy) | | |
|---------|------|------|--------|---------|-----|------|------|-----|------|------|
| | Visual | Textual | val(U) | test(U) | val | testA | testB | val | testA | testB |
| CRIS [40] | CLIP R101 | CLIP | 56.56 | 57.38 | 57.94 | 64.05 | 48.42 | 67.35 | 71.54 | 62.16 |
| VLT [15] | Swin-B | BERT | 63.49 | 66.22 | 63.53 | 68.43 | 56.92 | 72.96 | 75.96 | 69.60 |
| LAVT [7] | Swin-B | BERT | 61.24 | 62.09 | 62.14 | 68.38 | 55.10 | 72.73 | 75.82 | 68.79 |
| BKINet [41] | CLIP R101 | CLIP | 64.21 | 63.77 | 64.91 | 69.88 | 53.39 | 73.22 | 76.43 | 69.42 |
| ReLA [14] | Swin-B | BERT | 65.00 | 65.97 | 66.04 | 71.02 | 57.65 | 73.92 | 76.48 | 70.18 |
| SLViT [8] | SegNeXt-B | BERT | 62.75 | 63.57 | 64.07 | 69.28 | 56.14 | 74.02 | 76.91 | 70.62 |
| SADLR [42] | Swin-B | BERT | 63.60 | 63.56 | 62.48 | 69.09 | 55.19 | 74.24 | 76.25 | 70.06 |
| DMMI [43] | Swin-B | BERT | 63.46 | 64.19 | 63.98 | 69.73 | 57.03 | 74.13 | 77.13 | 70.16 |
| CGFormer [11] | Swin-B | BERT | 64.68 | 64.09 | 64.54 | 71.00 | 57.14 | 74.75 | 77.30 | 70.64 |
| RISCLIP [44] | CLIP-B | CLIP-B | 64.10 | 65.09 | 65.53 | 70.61 | 55.49 | 73.57 | 76.46 | 69.76 |
| MagNet [16] | Swin-B | BERT | 65.36 | 66.03 | 66.16 | 71.32 | 58.14 | 75.24 | 78.24 | 71.05 |
| LQMFormer [12] | Swin-B | BERT | 64.73 | 66.04 | 65.91 | 71.84 | 57.59 | 74.16 | 76.82 | 71.04 |
| ReMamber [9] | Mamba-B | CLIP | 63.90 | 64.00 | 65.00 | 70.78 | 57.53 | 74.54 | 76.74 | 70.89 |
| **SaFiRe** (Ours) | Swin-B | BERT | **67.07** | **66.87** | **66.39** | **71.96** | **58.47** | **75.34** | **78.52** | **71.35** |

**Traditional Benchmarks.** Tab. 1 presents a comparison of our **SaFiRe** with state-of-the-art methods on the RefCOCO, RefCOCO+, and RefCOCOg datasets with oIoU metric, where all models are trained independently on single datasets. Results show that **SaFiRe** outperforms all other methods across all datasets. In particular, **SaFiRe** achieved a great improvement on RefCOCOg, where the task involves more complex and fine-grained textual descriptions compared to RefCOCO.

In addition, Tab. 2 compares **SaFiRe** with recent state-of-the-art methods trained under mixed-data configuration. The precise dataset mixtures differ across methods. To avoid conflating performance with training-corpus choices, we report each baseline under its official training configuration. In general, these models draw on subsets and combinations of widely used referring-expression and segmentation datasets—such as RefCOCO [13], RefCOCO+ [13], RefCOCOg [37], COCO [38], Object365 [54], ADE20K [55], COCO-Stuff [56], PACO-LVIS [57], PASCAL-Part [58], GranD [20], VOC2010 [59], MUSE [53], gRefCOCO [14], and COCO-Interactive [52]. Our training corpus follows a mixed-data setup based on the RefCOCO series and COCO-Stuff, which is comparatively smaller. However, the results show that **SaFiRe** achieves competitive results on all three benchmarks, particularly excelling on the more challenging RefCOCOg and RefCOCO+ benchmarks. Notably, it surpasses several SAM-enabled models (e.g., u-LLaVA, GSVA) across multiple splits.

**aRefCOCO.** We further conduct a *zero-shot transfer evaluation* of different models on the aRefCOCO. The expressions in aRefCOCO are notably ambiguous and more complex, frequently lacking explicit category indications and containing more distractors, which demands a sophisticated cross-modal

Table 2: **Comparison of Model Performance Trained on Mixed Datasets on Three Benchmarks**. U indicates UMD partition of RefCOCOg. The best performances are in **bold**. (ft) denotes models further finetuned on RefCOCO/+/g after mix training. "Textual" and "Visual" refer to the textual encoder and the visual backbone, respectively. "LLM" denotes the large language model used as the textual reasoner.

| | Methods | Textual / LLM | Visual | RefCOCOg (hard) | | RefCOCO+ (medium) | | | RefCOCO (easy) | | |
|---|---|---|---|---|---|---|---|---|---|---|---|
| | | | | val(U) | test(U) | val | testA | testB | val | testA | testB |
| mIoU | EEVG [45] | BERT | Swin-B | 71.5 | 71.9 | 71.4 | 75.6 | 64.6 | 77.5 | 79.6 | 75.3 |
| | PromptRIS [46] | CLIP | SAM | 69.2 | 70.5 | 71.1 | 76.6 | 64.3 | 78.1 | 81.2 | 74.6 |
| | OneRef-B (ft) [47] | BEiT3-B | BEiT3-B | 74.1 | 74.9 | 74.7 | 77.9 | 69.6 | 79.8 | 81.9 | 77.0 |
| | C3VG [48] | BEiT3-B | BEiT3-B | 76.3 | 77.1 | 77.1 | 79.6 | 72.4 | 81.4 | 82.9 | 79.1 |
| | **SaFiRe** (Ours) | BERT | Swin-B | **77.3** | **78.1** | **77.4** | **80.5** | **72.9** | **82.1** | **83.7** | **80.1** |
| oIoU | MagNet [16] | BERT | Swin-B | 67.8 | 69.3 | 68.1 | 73.6 | 61.8 | 76.6 | 78.3 | 72.2 |
| | PolyFormer-L [49] | BERT | Swin-L | 69.2 | 70.2 | 69.3 | 74.6 | 61.9 | 76.0 | 78.3 | 73.3 |
| | UNINEXT-L [50] | BERT | ConvNeXt-L | 73.4 | 73.7 | 70.0 | 74.9 | 62.6 | 80.3 | 82.6 | 77.8 |
| | LISA (ft) [19] | LLaVA-7B | SAM | 67.9 | 70.6 | 65.1 | 70.8 | 58.1 | 74.9 | 79.1 | 72.3 |
| | GLaMM [20] | Vicuna-7B | SAM | 74.2 | 74.9 | 72.6 | 78.7 | 64.6 | 79.5 | 83.2 | 76.9 |
| | u-LLaVA [51] | Vicuna-7B | SAM | 74.8 | 75.6 | 72.2 | 76.6 | 66.8 | 80.4 | 82.7 | 77.8 |
| | PSALM*[52] | Phi-1.5 | Swin-B | 71.0 | 72.3 | 68.1 | 70.7 | 64.4 | 78.0 | 78.1 | 76.6 |
| | GSVA (ft) [21] | Vicuna-13B | SAM | 74.2 | 75.6 | 67.4 | 71.5 | 60.9 | 78.2 | 80.4 | 74.2 |
| | PixelLM [53] | Llama2-13B | CLIP ViT | 69.3 | 70.5 | 66.3 | 71.7 | 58.3 | 73.0 | 76.5 | 68.2 |
| | **SaFiRe** (Ours) | BERT | Swin-B | **75.8** | **76.8** | **74.8** | **78.9** | **68.3** | **81.4** | **83.6** | **78.2** |

* PSALM results are re-evaluated to align with standard settings [19], which predicts masks for each image-text pair for evaluation.

Table 3: **Performance Comparison on aRefCOCO.**

| Methods | aRefCOCO | | | | |
|---|---|---|---|---|---|
| | oIoU | mIoU | Pr@50 | Pr@70 | Pr@90 |
| LAVT [7] | 31.7 | 34.9 | 30.8 | 19.9 | 5.5 |
| CGFormer [11] | 54.0 | 60.3 | 67.0 | 59.7 | 27.2 |
| ReMamber [9] | 49.4 | 57.8 | 64.2 | 57.9 | 27.9 |
| MagNet [16] | 54.4 | 60.7 | 67.7 | 61.4 | 29.9 |
| EEVG [45] | 50.7 | 59.4 | 66.9 | 62.4 | 29.4 |
| C3VG [48] | 62.9 | 69.1 | 78.4 | 72.8 | 32.8 |
| LISA(7B) [19] | 49.0 | 53.8 | 58.7 | 49.2 | 20.6 |
| LISA++(7B) [60] | 45.3 | 50.4 | 51.5 | 43.5 | 19.7 |
| PSALM [52] | 60.1 | 67.3 | 75.7 | 71.8 | 38.7 |
| **SaFiRe** (Ours) | **65.1** | **71.4** | **78.9** | **73.6** | **45.2** |

Table 4: **Performance Comparison of Different Configurations.**

| Configuration | | RefCOCOg | | RefCOCO | |
|---|---|---|---|---|---|
| Saccade | Fixation | oIoU | mIoU | oIoU | mIoU |
| ✓ | ✓ | **67.07** | **69.68** | **75.34** | **77.02** |
| ✓ | | 65.72 | 68.24 | 74.27 | 76.12 |
| | ✓ | 66.32 | 68.73 | 73.60 | 75.64 |
| | | 64.48 | 67.72 | 72.92 | 75.17 |

understanding for precise segmentation. For fair comparison, all models have NOT been trained on aRefCOCO. As shown in Tab. 3, our model achieves the best overall performance on aRefCOCO, demonstrating its strong ability to precisely localize referred objects under ambiguous language conditions.

## 4.4 Ablation Study

**Module Ablation.** Tab. 4 summarizes the performance of different module configurations within our model. The results show that our full model achieves the highest oIoU scores on both RefCOCOg (descriptive sentences) and RefCOCO (concise phrases). Compared to independent module usage, the synergistic combination of Saccade and Fixation yields significant gains, highlighting their complementary effects. Removing both modules reduces the model to a naive baseline that simply concatenates textual and visual sequences for VSSM scanning, which performs worst among all variants, underscoring the essential role of both Saccade and Fixation operations.

**Fixation Window Size.** In Fixation operation, the approach is related to sequence arrangement for SSM scanning, where the ordering of input tokens impacts performance. With I representing individual image tokens and T representing the full sentence feature tokens, we conduct an ablation

Table 5: **Performance Comparison of Different Fixation Window Sizes.**

| Configuration | RefCOCOg |
|---|---|
| Vanilla [I...I T] | 64.38 |
| Repeat [I...I T...T] | 65.50 |
| Fixate $2 \times 2$ | 66.49 |
| **Fixate $4 \times 4$ (ours)** | **67.07** |
| Fixate $8 \times 8$ | 66.20 |

Table 6: **Performance Comparison of Different Backbones.**

| Backbone | Model | RefCOCOg |
|---|---|---|
| ViT-B | baseline | 55.55 |
| ViT-B | **SaFiRe** | 59.33 (**+3.78**) |
| Swin-B | **SaFiRe** | 67.07 |
| Swin-L | **SaFiRe** | 68.16 (**+1.09**) |

Table 7: **Comparison of Inference Efficiency.**

| Models | GFLOPs | FPS(image/s) |
|---|---|---|
| CGFormer [11] | 631 | 2.78 |
| PSALM [52] | 455 | 2.19 |
| LISA(7B) [19] | 4880 | 2.67 |
| LISA++(7B) [60] | 4943 | 1.29 |
| **SaFiRe** (Ours) | **384** | **8.26** |

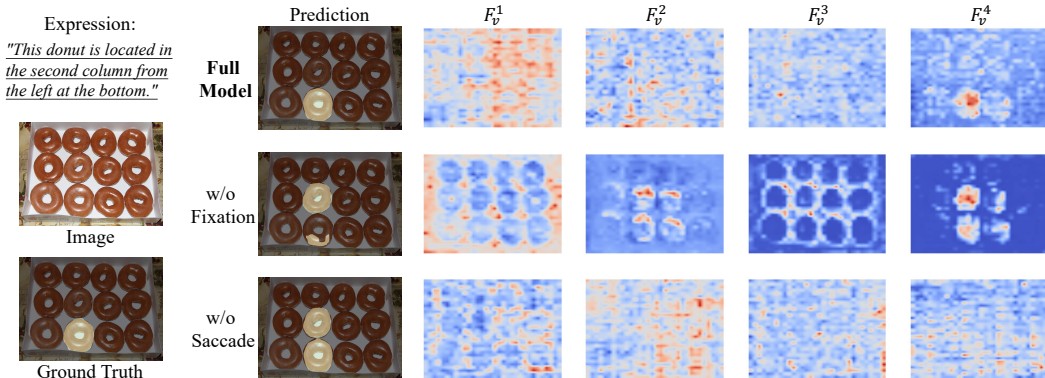

**Figure 3: Layer-Wise Visual Feature Maps.** Left→right corresponds to shallow→deep layers. The full model shows balanced activation. Without Fixation, local detail is missing; without Saccade, global focus weakens. The differing activation patterns reflect their complementary roles.

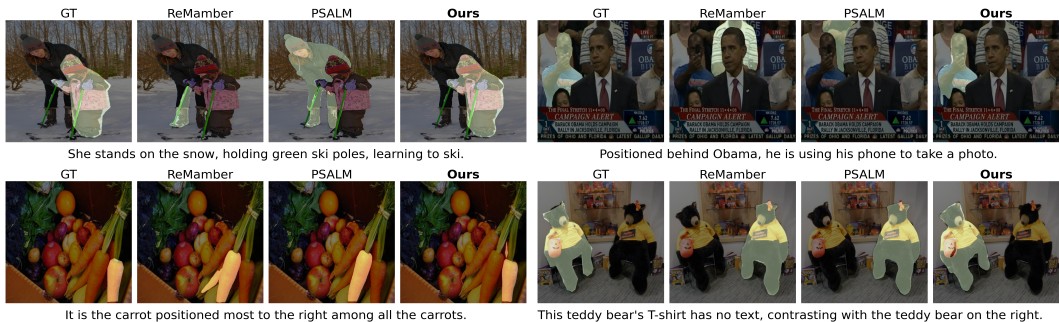

**Figure 4: Visualization Results for aRefCOCO.** Compared to the other two methods, our **SaFiRe** is more capable of comprehending ambiguous referring descriptions.

study of the following configurations in Tab. 5: (1) **Vanilla** `[I...I T]`: This configuration directly concatenates the image sequence `[I...I]` and the text sequence `T`. (2) **Repeat** `[I...I T...T]`: In this configuration, the text sequence is repeated multiple times at the end of the image sequence. (3) **Fixation** $x \times x$: This approach reiterates the text sequence once after every $x^2$ image tokens in window. For example, Fixation $2 \times 2$ means the sequence looks like: `[IIII T IIII T ....]`. Tab. 5 shows that the Repeat configuration improves upon Vanilla, but the Fixation operation, particularly with a window size of $4 \times 4$, shows a more significant enhancement in performance. This indicates that distributing textual information throughout the visual sequence helps maintain semantic continuity and guides the model to better align specific image regions with the text, rather than treating the two modalities as isolated blocks with simple concatenation.

**Backbone Ablation.** Here we present a comparison of different backbones, as depicted in Table Tab. 6. While our method is not specifically designed for global attention backbones such as ViT-B, it still brings a notable improvement of +3.78. Furthermore, when applied to stronger window-based backbones like Swin-L, the performance continues to improve, reaching 68.16 on RefCOCOg. These results indicate that our method generalizes well across different backbone architectures and particularly benefits from more powerful feature extractors.

**Layer-wise Feature Map Analysis.** In Fig. 3, we visualize the output visual feature maps of each **SaFiRe** layer under different model configurations. A clear difference in **activation patterns** can be observed between the Saccade and Fixation operations. Specifically, the Saccade operation generates broader and more globally distributed activations that align with the overall shape and location described in the text. In contrast, the Fixation operation produces sharper, more localized activations, focusing on fine-grained visual details. When used independently, each module exhibits limitations in fully capturing and grounding complex language expressions. However, when combined, their complementary activation behaviors allow the model to achieve an accurate identification of the

described object. This further confirms that Saccade and Fixation work in synergy to enhance the model's cross-modal understanding.

## 4.5 Inference Efficiency Comparison

We evaluate inference efficiency under identical input settings and report both computational cost (GFLOPs) and throughput (FPS). As shown in Tab. 7, **SaFiRe** is compared against representative transformer-based (CGFormer [11]) and LLM-driven models (PSALM [52], LISA [19], LISA++ [60]). **SaFiRe** attains 8.26 FPS with 384 GFLOPs, yielding the highest throughput and the lowest compute among all compared methods. Together with the accuracy results in Tab. 1, Tab. 2 and Tab. 3, these findings underscore the efficiency advantage of our framework.

## 4.6 Visualization Results

Fig. 4 compares **SaFiRe** with two state-of-the-art RIS approaches—Mamba-based ReMamber [9] and LLM-driven PSALM [52]—on the **aRefCOCO** benchmark. Our method exhibits a stronger ability to handle referential ambiguity and avoids the confusion observed in the other two methods. For additional examples, see Appendix E.

## 4.7 Failure Case Analysis

GT                        Prediction                              GT                        Prediction

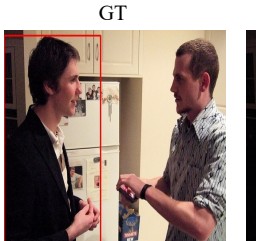 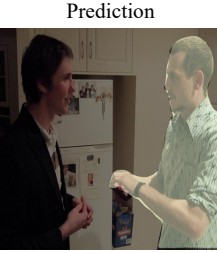 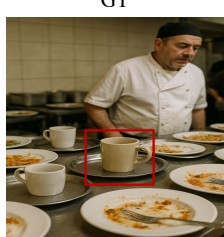 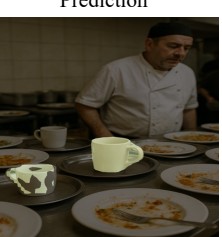

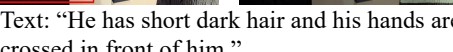

Text: "He has short dark hair and his hands are crossed in front of him."

Text: "the mug on the tray closer to the plate nearest the chef."

(a) Extremely similar visual semantics

(b) Unusually long relational chain

Figure 5: **Two Typical Types of Failure Cases.**

To guide future works, we analyze two typical failure scenarios. (1) Extremely similar visual semantics. Errors arise when multiple candidates share similar visual semantics but only one best fits the intended meaning. As in Fig. 5a, both people look "cross-like" at first glance; zooming in reveals that one is actually holding an object rather than truly crossing his hands. These cases demand particularly fine-grained visual discrimination and action-semantic grounding, which the model overlooked. (2) Unusually long relational chain. Failures occur in texts with very long spatial or relational chains. These require multi-step reasoning with frequent spatial shifts, where errors in intermediate links accumulate, as shown in Fig. 5b.

## 5 Conclusion

We present **SaFiRe**, a cognitively inspired framework for referring image segmentation that addresses the limitations of existing keyword-matching approaches. Motivated by the challenge of referential ambiguity, our method simulates the human two-phase grounding process through alternating *Saccade* and *Fixation* operations. Built upon the Mamba state-space model, **SaFiRe** enables efficient global-to-local semantic alignment and achieves state-of-the-art performance on both standard and ambiguous datasets, including the newly proposed aRefCOCO.

**Limitation and Future Work.** Despite its effectiveness, **SaFiRe** still faces challenges in handling language diversity in real-world application, which cannot be fully represented by existing datasets. Future work will explore incorporating structured scene understanding and pragmatic reasoning, and extending the framework to dialog-based and video grounding tasks. Moreover, **SaFiRe** has the potential to be adapted to other tasks or modalities with minimal modification, such as Open-Vocabulary Segmentation (OVS) [61–63] by adjusting the task head, or Audio-Visual Segmentation (AVS) [64–66] by substituting the backbone.

## Acknowledgments

This work is supported by the National Key R&D Program of China (No. 2022ZD0160702), National Natural Science Foundation of China (No. 62306178) and STCSM (No. 22DZ2229005), 111 plan (No. BP0719010).

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

# A  Limitations

While our method achieves strong performance across various benchmarks, several limitations must be acknowledged. First, the model's ability to process extremely complex or ambiguous expressions remains limited. Although we evaluated it primarily on datasets like RefCOCO family and aRefCOCO, these datasets may not fully represent the diversity of real-world scenarios. Additionally, for dataset construction, we rely on a large language model to generate textual descriptions. While this approach ensures a broad range of expressions, it may inadvertently introduce bias, even after meticulous filtering.

# B  Broader Impacts

The work presented here has the potential for both positive and negative societal impacts. On the positive side, our method's ability to understand complex referring expressions and improve image segmentation could significantly benefit fields such as medical imaging, robotics, and accessibility technologies—particularly for individuals with disabilities. On the negative side, the model may inherit biases from pre-trained encoders (e.g., Swin or BERT), potentially leading to stereotypical associations in referring expressions (e.g., linking gender or social roles to certain objects), thereby amplifying societal biases in segmentation outcomes.

# C  More Related Work

**From Traditional Segmentation to Language-Based Segmentation.** Traditional segmentation has long focused on delineating object regions from visual cues alone, employing various methods such as CNN [67–72], Vision Transformers [73–78], and Diffusion Models [79–86]. Moving beyond purely visual cues, **language-based segmentation** introduces linguistic grounding, enabling segmentation guided by natural language. Within this field, *referring image segmentation* (RIS) aims to localize and segment objects described by textual expressions, yet many methods still treat it as a shallow keyword-matching task, leaving ambiguous expressions unresolved. *Open-vocabulary segmentation* (OVS), in contrast, emphasizes category generalization and broad semantic coverage through large-scale vision–language pretraining [61–63, 87–90]. More recently, *reasoning segmentation*, introduced by LISA [19] extends beyond RIS by requiring complex reasoning or world knowledge, leveraging large language models to perform such inference. Our work, however, remains within the RIS scope: rather than relying on external knowledge, we focus on resolving the referential ambiguities that an ideal RIS model *should* handle but current approaches *fail* to address.

**Mamba and Its Broader Applications.** Recently, **Mamba** has been actively explored across unimodal and multimodal learning [91–93], owing to its ability to model long-range dependencies with linear-time efficiency. It has since been adopted as a general-purpose backbone in diverse downstream tasks. In the visual domain, Mamba has shown promise in video understanding and spatiotemporal modeling [94, 95], where its state-space formulation enables efficient temporal integration while preserving spatial fidelity. Beyond vision, it has been applied to audio and speech modeling [96], biomedical image analysis [97], and robotics with vision–language–action models [98], demonstrating strong adaptability across modalities. These developments highlight Mamba's growing influence as a unified sequence modeling paradigm. In this work, we further leverage its long-context sequential modeling capacity and scan-then-update dynamics—mirroring the saccade–fixation process of human perception—to resolve referential ambiguities in referring image segmentation.

# D  More Details of aRefCOCO Dataset

## D.1  Generation, Filtering and Validation

### D.1.1  Generation Prompts

In order to ensure the richness and diversity of object descriptions while adhering to the fundamental requirement of single-object localization in RIS tasks, we used a prompt template for Qwen-VL-Max as shown in Appendix D.1.1.

```
### Role
You are a professional assistant, proficient in object recognition and
description. Your task is to provide a clear and focused description of the main
object inside the red box in the image. You are given the category information of
the main object inside the red box as '{target}'. The description should be based
on the provided target information, and you should generate a more detailed
description with your visual analysis of the image. Please note that there is no
visible box in the actual image.

### Target Category Information
targt category information: '{target}'

### Skills
1. [Important!] Please only focus on the main object, i.e., '{target}', and
ignore the background or irrelevant details.
2. [Important!] Do not assume or include any information that is not present in
the image content.
3. [Important!] Provide 5 independent English sentences, each of which can
independently identify '{target}' without relying on other sentences. Each
sentence should describe a unique identifying feature.
4. [Important!] Use the subject '{target}' from the targt category information,
and ensure you are describing a single object '{target}'! Others should be able
to locate only one object based on your sentences.
5. [Important!] Use singular language to ensure the description focuses only on
one main object. Do not use plural words like 'they' as the subject, but you are
allowed to use pronouns as the subject.
6. [Important!] If you need to describe based on objects around '{target}',
please keep the subject as '{target}'.
```

### D.1.2 Filtering

We first pre-screen all descriptions with GPT-4o for the two patterns:"object-distracting" (i.e., containing multiple nouns) and "category-implicit" (i.e., where the subject is a pronoun). Items that match either pattern are retained. Then, to verify the remaining pairs meet the basic rules we set for annotation, GPT-4o is involved as an evaluator. The prompt is as shown in Appendix D.1.2.

Filtering Prompts

```
Role:
You are a professional visual-language evaluator. Your task is to verify whether
each given description sentence is a faithful, grounded, and rule-abiding
description of the target object **inside the red box** in the provided image.

Information:
The target object belongs to the category: {target}.

The description was generated under the following constraints:
1. It must describe only the main object ('{target}'), ignoring the background or
other irrelevant elements.
2. It must not include objects that are not in the whole image, but objects out
of box are allowed.
3. Each sentence must independently identify the target.
4. It can contains objects beyond the target.
5. It must use singular phrasing, and refer to the object using either its
category label ('{target}') or singular pronouns.

You will be given 5 sentences. Your task is to verify, for each sentence, whether
it satisfies **all** the above requirements based solely on the image content.

Sentences to Verify:
{Sentences}
```

```
Instructions:
1. For each sentence, respond with **"yes"** or **"no"**. if "no", follow it by a
**brief reason**.
2. If any rule is violated or the content is ungrounded, answer "no" and specify
which rule(s) are broken.
3. Do not use external knowledge or assumptions. Base your judgment strictly on
what is visible in the image.
4. Provide one answer per sentence, in order. Each answer must be on its own
line.
5. Do not add commentary outside the answers. Return only the answers.
6. Ensure the number of answers equals 5.
```

### D.1.3 Validation

Manual review was conducted by at least two annotators with RIS domain expertise. Each pair was independently validated to confirm (1) whether the description uniquely refers to a single object in the image, and (2) whether the expression obeys our referential ambiguity criteria while remaining image-groundable. In borderline cases, annotators reached consensus through discussion, and a third reviewer was consulted when needed.

### D.2 Comparison with Original Descriptions

We conduct quantitative comparisons to highlight the distinct features of aRefCOCO compared to the original descriptions, as visualized in Fig. 6. Specifically, the comparisons focus on the following aspects:

**Description Length.** As shown in Fig. 6a(i), compared to the original descriptions in RefCOCO and RefCOCOg, aRefCOCO contains a higher proportion of longer sentences and significantly reduces the frequency of short, phrase-like descriptions. This shift towards more complex sentence structures underscores the value of aRefCOCO as an extension to the RefCOCO series, making it particularly relevant for real-world RIS tasks that involve longer and more intricate sentence constructions.

**Relational Words.** We calculated the growth rates of the most frequent words in the original and aRefCOCO descriptions, focusing on those strongly correlated with the relationships between targets and distractors. The results in Fig. 6a(ii) show a significant increase in these words, particularly prepositions like "and", "to", and "of", which grew by 450.7%, 288.6%, and 249.8%, respectively. These growths contribute to more structurally and semantically complex sentences, enhancing spatial and relational expressions and enriching the semantic content in aRefCOCO descriptions.

**Text Embedding Distribution.** To further examine the semantic differences between the original and aRefCOCO descriptions, we visualize their text embeddings in a reduced 3D space using PCA, as shown in Fig. 6b. The figure illustrates a clear shift in embedding distributions, where the aRefCOCO (red) embeddings are consistently displaced from their original (blue) counterparts. This systematic divergence indicates that the rewritten descriptions introduce substantial semantic variations rather than minor lexical edits.

## E More Visualization Results

In this section, we provide more visualization results of our **SaFiRe**, ReMamber [9] and PSALM [52] on aRefCOCO for further comparison, as shown in Fig. 7.

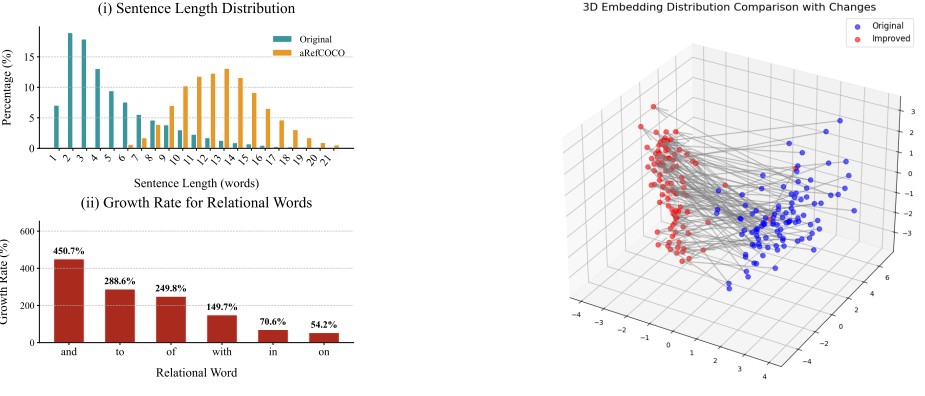

(a) Sentence-level statistics comparison   (b) 3D text embedding distribution comparison

Figure 6: **Comparison of Sentence-Level Statistics and Text Embedding Distributions Between the Original Descriptions and aRefCOCO.**

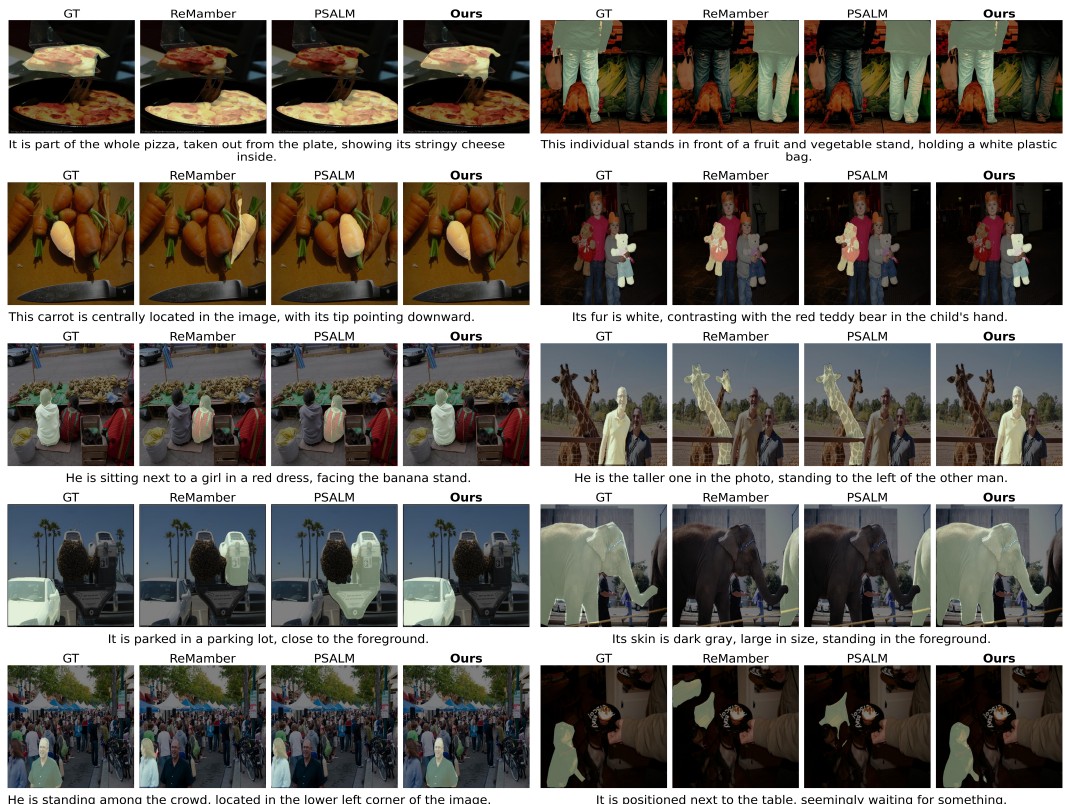

Figure 7: **More Results on aRefCOCO (Part 1).**

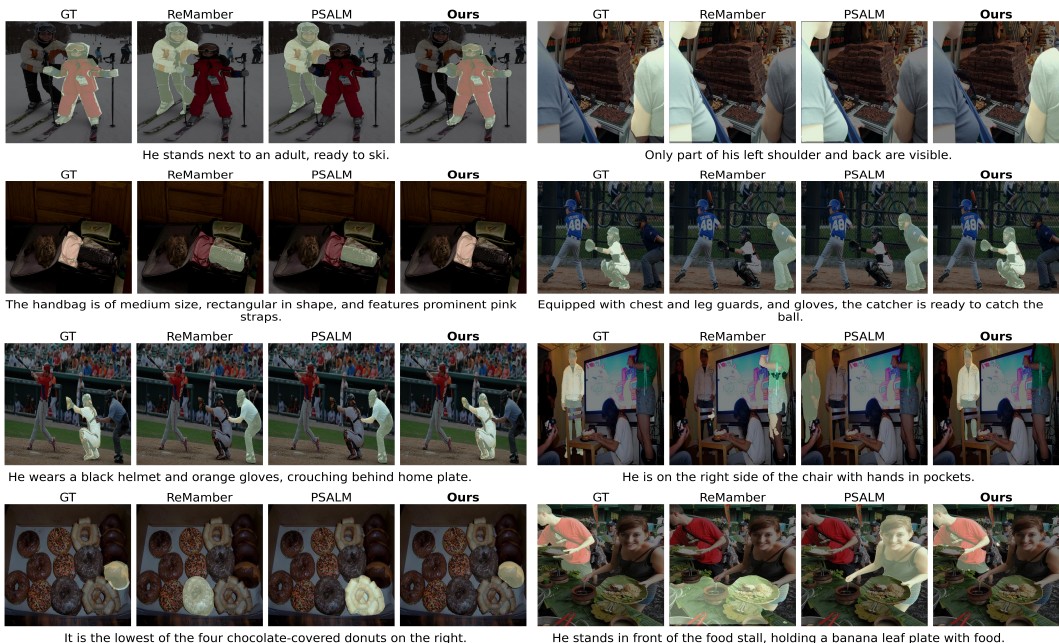

Figure 7: **More Results on aRefCOCO (Part 2, continued).**

