# OpenReview forum: "SaFiRe: Saccade-Fixation Reiteration with Mamba for Referring Image Segmentation"
_NeurIPS.cc/2025/Conference — NeurIPS 2025 poster_

### Official Review · Reviewer_R7DT · 2025-06-23

**Clarity:** 3
**Significance:** 3
**Originality:** 2
**Rating:** 4
**Confidence:** 4

**Summary:**

This paper critiques existing approaches in Referring Image Segmentation (RIS) for their reliance on simple keyword matching, which often fails in ambiguous situations. To address this limitation, the paper introduces a framework called SaFiRe, inspired by human cognitive processes consisting of a global understanding phase (Saccade) followed by a fine-grained refinement phase (Fixation). The architecture utilizes the Mamba state-space model to effectively manage long-range dependencies, aligning with the cognitive scan-then-inspect paradigm. Furthermore, to evaluate model performance under referential ambiguity, the paper presents a new benchmark, aRefCOCO, specifically designed to include challenging cases such as object-distracting and category-implicit expressions. The proposed method consistently outperforms existing approaches across both standard and ambiguous benchmarks.

**Questions:**

1. Novelty Clarification: How does SaFiRe fundamentally differ from “Reasoning Segmentation” approaches such as LISA? Is the distinction only architectural, or is there a deeper conceptual or functional divergence?

2. Failure Case Insights: While ambiguous expressions are addressed, what patterns emerge in failure cases (e.g., pronoun chains, conflicting attributes)? Are there categories of ambiguity where SaFiRe still struggles?

3. Reasoning Model Benchmarking: Have the authors considered evaluating SaFiRe against approaches that share similar conceptual (i.e., reasoning-centric baselines like LISA or LISA++)? A clearer empirical comparison could help position this work more accurately within the reasoning segmentation landscape.

**Ethical Concerns:**

["NO or VERY MINOR ethics concerns only"]

**Final Justification:**

The rebuttal addresses most of my concerns.

**Limitations:**

YES.
The authors have adequately outlined both the methodological and ethical limitations of their work. A suggestion to enhance this section would be to include failure case visualizations, which would complement the limitations by providing concrete examples along with appropriate explanations. This addition could help illustrate the specific challenges encountered and further clarify the contexts in which the proposed method may struggle.

**Quality:**

3

**Strengths And Weaknesses:**

The paper makes an intriguing contribution to the field, with current strengths and potential suggestions outlined below:
Strengths:
1. Methodological Novelty: The concept of mapping the two-phase human cognition-inspired “Saccade” for global coarse grounding and “Fixation” for local refinement provides an interesting and well-conceived framework for addressing RIS challenges.
2. Architectural Efficiency: Leveraging the SFLayer in existing segmentation pipelines through Mamba’s linear complexity enables scalable refinement, which is reflected in improved inference speed during evaluations.
3. New Benchmark: The identification of oversimplified referring expressions in existing RIS benchmarks and the creation of a dataset focused on evaluating model performance under ambiguous expressions represent a significant advancement in addressing real-world linguistic challenges.
4. Evaluations and Experimentations: A clear evaluations across various methods on common benchmarks, along with clear ablations conducted in the paper, enhance understanding of the proposed method's advantages and the components contributing to its results.
Weakness:
1. Problem Novelty: The problem addressed in this paper closely resembles that of one of the cited papers (LISA), with insufficient articulation of the differences between the two. Clearly distinguishing between these works is crucial to avoid confusion regarding their focus on “reasoning segmentation” under ambiguous language prompts.
2. Underexplored comparisons: Given the objective of handling ambiguity in text prompts for segmentation, it is essential to explicitly discuss and evaluate reasoning models like LISA and LISA++ to clarify SaFiRe’s position within the wider community.
3. Benchmark: The availability of aRefCOCO as a test-only dataset could limit its utility within the community. Providing a clear train/val split would enhance its usefulness for future benchmarking efforts.
4. Quantification of Speedup: While the proposed method's speedup is specified, quantifying it in relation to computational requirements (GFLOPS) would be beneficial for the community.

---

> ### Author Rebuttal · Authors · 2025-07-31
>
> Thank you for the thoughtful review. We would like to first thank you for identifying the methodological novelty and efficiency of our Saccade–Fixation design—an assessment that echoes 8mtA’s praise for its robust alignment and jD3V’s recognition of its accuracy and speed, while Wg1a also noted the results to be promising.
>
> Below, we address each of your comments point‑by‑point and clarify how our revisions and additional experiments resolve the concerns you raised.
>
> >**R7DT-W1 && R7DT-Q1:** What are the fundamental differences between SaFiRe and reasoning segmentation approaches like LISA?
>
> **R7DT-W1-A && R7DT-Q1-A:**
> We first emphasize that SaFiRe and LISA tackle **fundamentally different tasks**.
> LISA brings up a novel task beyond RIS, while our SaFiRe focuses on bringing current RIS task into a more practical usage. We compare **Existing RIS**, our persuade on **Ideal RIS** and **ReasonSeg** as the following table.
>
> ||Existing RIS|Ideal RIS|ReasonSeg|
> |-|-|-|-|
> |Segment target based on text| v|v|v|
> |Simple expression|v|v|v|
> |**Ambiguous expression**|-|**v**|-|
> |Complex reasoning|-|-|v|
> |World knowledge|-|-|v|
>
> In few words: **(1)** LISA focus on what other RIS models **can NOT** do, while **(2)** we focus on what other RIS models **should do but fail**.
>
> ### (1) LISA focus on what other RIS models can NOT do.
>
> The ReasonSeg task proposed in LISA aims to response queries requiring **complex reasoning or world knowledge**. However, **complex reasoning** is an ability emerged from LLMs' large corpus pertraining; while **world knowledge** also requires large diversity of training data and large parameter size for memorize. LISA utilizes LLaVA (7B) to achieve this. Obviously, these abilities are out of the ability of a standard RIS architecture (typically with Swin+Bert as backbone).
>
> ### (2) We focus on what other models should do but fail.
>
> We focus on the referential ambiguity problem in RIS task: dealing with **object-distracting** or **category-implicit** expression. The cases in our aRefCOCO are still **in RIS’s scope** and an ideal RIS model should complete these cases perfectly. However, **existing RIS models tends to fail** due to their inherent limitations: they models RIS as a "key word matching problem". Our work shows that by **jumping out of "key word matching" paradigm**, we’re able to solve these hard cases.
>
> As for LISA, through it brings novel idea and solution about reasoning ability, they still fails on our **"should do but fail"** cases. Since they utilize SAM-style architecture for mask producing, which is still inside “key word matching” paradigm.
>
> ### (3) Evaluating LISA on our aRefCOCO benchmark.
>
> Here we provide the evaluation results of LISA and LISA++ on aRefCOCO. Notably, while LISA++ is designed to outperform LISA on ReasonSeg tasks, it actually performs worse than LISA on aRefCOCO.
> This indicates that, pursuing stronger reasoning ability may harm the model's performance on ambiguous expression, further highlights the necessity of introducing such a benchmark.
>
> | Method | oIoU | mIoU | Pr@50 | Pr@70 | Pr@90 |
> |--------|------|------|-------|-------|-------|
> | LISA(7B) | 49.0 | 53.8 | 58.7 | 49.2 | 20.6 |
> | LISA++(7B) | 45.3 | 50.4 | 51.5 | 43.5 | 19.7 |
> | **SaFiRe (Ours)** | **65.1** | **71.4** | **78.9** | **73.6** | **45.2** |
>
> We appreciate the reviewer's comments and will add above analysis to the revision for better clarity.
>
>
> >**R7DT-Q3:** Benchmarking SaFiRe on the Reasoning Segmentation task
>
> **R7DT-Q3-A:**  As explained above, SaFiRe is not designed for reasoning segmentation, we believe it is unfair and unreasonable to evaluate SaFiRe on the ReasonSeg benchmark, as it does not have access to world knowledge.
>
>
> >**R7DT-Q2:** Failure Case Insights: While ambiguous expressions are addressed, what patterns emerge in failure cases (e.g., pronoun chains, conflicting attributes)? Are there categories of ambiguity where SaFiRe still struggles?
>
>
> **R7DT-Q2-A:** Thank you for your question. We will add detailed visualization of failure cases in revised version. Here, we summarize below the main patterns observed in SaFiRe’s failure cases:
> - **Fine-grained attribute ambiguity.** Failures often occur when multiple candidates share broadly similar attributes but only one matches the intended semantics of the description. In such cases, even subtle differences in attributes (e.g., hand positioning) can carry decisive referential meaning, yet the model may overlook them due to visual similarity.
> *Example: “He crosses his hands while talking to the other person.”* In the image, one man rests one hand over the other, while another interlaces his fingers. Although both are hand-related gestures and might visually resemble “crossed hands,” only the latter accurately captures the intended meaning. The model, however, selects the former, revealing its difficulty in grounding fine-grained action attributes to textual descriptions.
>
> - **Long spatial/relational chain ambiguity.** Another common failure mode arises in texts involving long, compositional chains of spatial or relational references. These expressions require the model to perform multi-step understanding with frequent spatial shifts, jumping between different objects across the scene. A typical symptom is that the grounding process breaks down at one or more intermediate links, leading to cumulative error.
> *Example: “the mug on the tray beside the plate nearest the chef.”* In the image, there are multiple mugs, trays, and plates spread across a crowded table, and the chef is positioned off to one side. There is no obvious shortcut to the target—only by following the full relational chain can the correct object be identified. The model, however, misidentifies an object linked to an incorrect plate early in the chain, revealing its difficulty in maintaining spatial consistency and grounding excessive relational dependencies to textual descriptions.
>
> We will add the above analysis into revision to clarify the potential future directions.
>
> >**R7DT-W3:** Benchmark: The availability of aRefCOCO as a test-only dataset could limit its utility within the community. Providing a clear train/val split would enhance its usefulness for future benchmarking efforts.
>
> **R7DT-W3-A:** Thank you for your suggestion. The primary goal of our paper is to provide a robust and challenging benchmark for evaluating model performance under referential ambiguity, rather than to construct a large-scale training dataset. We fully agree that establishing a training split for aRefCOCO could further benefit the community and enhance future research. We will consider construct train/val splits in the future.
>
> >**R7DT-W4:** Quantification of Speedup: While the proposed method's speedup is specified, quantifying it in relation to computational requirements (GFLOPS) would be beneficial for the community.
>
> **R7DT-W4-A:** Thank you for your valuable suggestion. We have added GFLOPs results, including both LISA and LISA++ in our comparison. Furthermore, we present the speed comparison in terms of FPS (image-text pair/s).  As shown in the table below, SaFiRe achieves a much lower computational cost and faster inference speed compared to LISA and LISA++, which both require near 5000 GFLOPs (SAM- and LLM-driven) and are much slower. At the same time, SaFiRe delivers significantly better segmentation accuracy on the RIS task.
>
> | Model           | GFLOPs | FPS   |
> |-----------------|--------|-------|
> | CGFormer        | 631    | 2.78  |
> | PSALM           | 455    | 2.19  |
> | LISA-7B         | 4880   | 2.67  |
> | LISA++(7B)      | 4943   | 1.29  |
> | **SaFiRe (Ours)** | **384**    | **8.26** |

---

> > ### Comment · Reviewer_R7DT · 2025-08-05
> >
> > I thank the authors for the detailed response. Their reply addresses most of my concerns, and I will update my rating positively to reflect this.

---

> > > ### Author Response · Authors · 2025-08-06
> > > **Thank you for the response**
> > >
> > > Dear Reviewer R7DT,
> > >
> > > Thank you very much for taking the time to review our submission and for your constructive feedback. We truly value your suggestions and will incorporate the relevant points you raised into the final version of the paper. We greatly appreciate your efforts in helping us strengthen our work.
> > >
> > > Best regards,
> > >
> > > Authors of Paper 2533

---

### Official Review · Reviewer_Wg1a · 2025-06-30

**Clarity:** 3
**Significance:** 3
**Originality:** 2
**Rating:** 4
**Confidence:** 5

**Summary:**

The paper presents a novel framework, called SaFiRe, to address the challenges in referring image segmentation: object-distracting expressions and category-implicit expressions. It is based on a Mamba’s scan-then-update function. A new benchmark called aRefCOCO is also introduced for evaluating on ambiguous expressions. Experimental results are promising when compared to existing RIS methods.

**Questions:**

Please refer to the weaknesses.

**Ethical Concerns:**

["NO or VERY MINOR ethics concerns only"]

**Final Justification:**

I appreciate the detailed response provided by the authors. The rebuttal addresses most of my concerns. I will maintain my initial score.

**Limitations:**

yes

**Quality:**

3

**Strengths And Weaknesses:**

Strengths:
+ Experiment results are promising.
+ The utilization of state-space sequence modeling for RIS is reasonable.

Weaknesses:
- The motivation of the paper is kind of vague and the explanation of SaFire’s design is questionable. Even though the authors claim that their `Saccade` and `Fixation` come from cognitive science, it can not convince me that such operation aligns with their original aspects.
- Also, the novelty of the proposed framework is questionable, as it borrows largely similar designs from VMamba, like the scanning and folding techniques in other SSM works.
- It is unclear whether the samples selected for the aRefCOCO are valid or not when including VLM (i.e., Qwen-VL-Max) for curation due to the absence of further discussion of it. This would also cause a potential cherry-picking by selecting challenging samples from RefCOCO where the proposed method outperforms and the baselines do not. A good practice is to provide more samples in the Supp. Materials for evaluation.

---

> ### Author Rebuttal · Authors · 2025-07-31
>
> Thank you for your time on our submission. We would like to first thank you for acknowledging the promising experimental results and the suitability of state‑space modeling for RIS—results similarly lauded by 8mtA and jD3V, with R7DT additionally emphasizing the methodological novelty behind them.
>
> Below, we address each of your comments point‑by‑point and clarify how our revisions and additional experiments resolve the concerns you raised.
>
> >**Wg1a-W1:** The motivation of the paper is kind of vague and the explanation of SaFire’s design is questionable. Even though the authors claim that their Saccade and Fixation come from cognitive science, it can not convince me that such operation aligns with their original aspects.
>
> **Wg1a-W1-A**:
> First, we would like to clarify that our main goal is to resolve **referential ambiguity**, a fundamental challenge in real-world RIS scenarios. Previous methods struggle in such scenarios due to their reliance on keyword-matching modeling. Instead, we jump out of such paradigm and propose our Saccade-Fixation mechanism to address the issue.
>
> Second, we are inspired, rather than conducting a bio-faithful simulation to the original concepts. Our Saccade–Fixation mechanism is a computational **abstraction**: Saccade performs global scanning for coarse understanding , while Fixation executes focused, token-level binding and language alignment.
>
> Additionally, our motivation has also been recognized by Reviewer jD3V as "**well-motivated and grounded in practical RIS limitations**" and by Reviewer R7DT as "**well-conceived**."
>
> Further, as already cited in our paper, two well-established findings ([17], [18]) support the scan–then–inspect pattern that inspires our approach. Below, we provide a detailed table that includes direct citations, explanations, and a mapping of each theory to its concrete implementation in SaFiRe, showing how we are inspired.
>
> | Classical Finding                                                                                                   | Key Quotation                                                                                                                                                                                                                      | Implementation in SaFiRe                                                                                                                                                                   |
> |---------------------------------------------------------------------------------------------------------------------|------------------------------------------------------------------------------------------------------------------------------------------------------------------------------------------------------------------------------------|--------------------------------------------------------------------------------------------------------------------------------------------------------------------------------------------|
> | **Feature-Integration Theory** (Treisman & Gelade, 1980): Features are registered in parallel, but *objects* emerge only when focal attention serially “fixes” each location. | “Features are registered early, automatically, and in parallel… stimulus locations are processed **serially with focal attention**. Any features present in the same central **fixation** are combined to form a single object.”   | ***Saccade***: Serial sweep over patches to supply coarse candidates; ***Fixation***: binds local visual tokens to the text, providing the “glue” Treisman describes. |
> | **Sentence–Picture Verification Model** (Clark & Chase, 1972): Comparing language to images proceeds through “encode → locate → verify” sub-operations whose latencies add up. | “Sentences and pictures are compared in an **algorithmic series of mental operations, each of which contributes additively to the response latency**; encoding, comparing and responding are serially ordered stages.”             | ***Saccade***: Supplies the encoded, coarse alignment (encode + locate); ***Fixation***: Iteratively verifies and refines those candidates, mirroring the verification stage.                    |
>
> We appreciate the reviewer's comments and will enrich the introduction part to better clarify our motivation in our technical design.
>
> >**Wg1a-W2:** The novelty of the proposed framework is questionable, as it borrows largely similar designs from VMamba, like the scanning and folding techniques in other SSM works.
>
> **Wg1a-W2-A**: We believe there are some misunderstandings.
>
> VMamba is designed for **single-modal vision tasks**. We use the SS2D block serves as a **fundamental processing unit** in our framework, similar to how attention mechanism functions in other architectures. Our novelty lies in the tailored design of Saccade-Fixation mechanism for tackling **multimodal RIS tasks**: First, in Saccade operation, the visual features are **modulated** by the overall meaning of the text and then are scanned to grasps a quick glimpse of image-text correspondence; Second, the Fixation operation follows a **carefully designed "group-scan-recover"** pattern to achive detailed examination of cross-modal alignment. We have conducted comprehensive ablation studies (see Tables 4 and 5) and visual comparisons (see Fig. 3) to validate the effectiveness of our approach.
>
> Additionally, we clarify that our goal is to better address the RIS task in referial ambiguity, rather than merely designing a new version of Mamba architecture.
>
> >**Wg1a-W3:** It is unclear whether the samples selected for the aRefCOCO are valid or not when including VLM (i.e., Qwen-VL-Max) for curation due to the absence of further discussion of it. This would also cause a potential cherry-picking by selecting challenging samples from RefCOCO where the proposed method outperforms and the baselines do not.
>
>  **Wg1a-W3-A**: Sorry for any confusion. The original samples from RefCOCO are randomly selected and re-annotated using Qwen via its API. Our SaFiRe model is **NOT** involved during the entire dataset construct process. For more details about the dataset construction, please refer to **8mtA-W2-A** to Reviewer 8mtA. We will release the code and dataset upon publication.

---

> > ### Comment · Reviewer_Wg1a · 2025-08-06
> >
> > I appreciate the detailed response provided by the authors. The rebuttal addresses most of my concerns. I will maintain my initial score.

---

> > > ### Author Response · Authors · 2025-08-06
> > > **Thank you for the response**
> > >
> > > Dear Reviewer Wg1a,
> > >
> > > Thank you for your review and thoughtful feedback. We are pleased that our rebuttal has addressed most of your concerns. We will incorporate your suggestions and include the relevant discussion in the final version of the paper.
> > >
> > > Best regards,
> > >
> > > Authors of Paper 2533

---

### Official Review · Reviewer_jD3V · 2025-07-02

**Clarity:** 3
**Significance:** 2
**Originality:** 3
**Rating:** 4
**Confidence:** 4

**Summary:**

This paper proposes a referring image segmentation framework that tackles two challenges: object-distracting expressions and category-implicit expressions. Built upon Mamba, the model first forms a global understanding and then refines it through region-wise inspection. A new benchmark is introduced to evaluate RIS models under ambiguous settings. Experiments show the effectiveness of the proposed method.

**Questions:**

1. How is the inference speed compared to ReMamber [9]?
2. What are the failure cases of the proposed method?

**Ethical Concerns:**

["NO or VERY MINOR ethics concerns only"]

**Final Justification:**

The rebuttal addresses most of my concerns. I will maintain my initial score.

**Limitations:**

Yes

**Quality:**

2

**Strengths And Weaknesses:**

[Strengths]
1. The paper is overall well-written and easy to follow.
2. The identification of object-distracting and category-implicit expression types is well-motivated and grounded in practical RIS limitations.
3. The introduction of the new benchmark evaluates RIS models under ambiguous referring expressions, which better reflects real-world challenges.
4. The experiments show the superiority of the proposed method in accuracy and inference speed.

[Weaknesses]
1. While the Saccade-Fixation design is novel, the rest of the model relies heavily on standard components (Swin-B, BERT). The innovation is primarily in the fusion mechanism rather than in the overall pipeline, which may limit its extensibility to other tasks or modalities without further architectural adjustments.

2. Although the introduced benchmark aRefCOCO increases ambiguity, it is still generated by a VLM, which could inadvertently introduce biases, even with meticulous filtering. The paper lacks evaluations on real-world, human-annotated ambiguous datasets or interactive grounding scenarios, limiting its external validity.

3. The paper could benefit from more analysis on failure cases of the proposed method. This would help clarify its limitations, especially in handling complex descriptions.

---

> ### Author Rebuttal · Authors · 2025-07-31
>
> Thank you for the careful review. We would like to first thank you for valuing the paper’s clarity and our method’s novelty，accuracy and speed—strengths also stressed by R7DT and supported by the promising results noted by Wg1a, while 8mtA further commended the strong gains on ambiguous benchmarks.
>
> Below, we address each of your comments point‑by‑point and clarify how our revisions and additional experiments resolve the concerns you raised.
>
> >**jD3V-Q1:** How is the inference speed compared to ReMamber?
>
> **jD3V-Q1-A:** We have added a inference speed comparison (by fps) with ReMamber as follows. Our model achieves  **1.46×** speedup over ReMamber. That's because its Twister design involves two operations with relatively large-scale computation. As a result, although we use Swin transformer (vs ReMamber's Mamba-B) as visual backbone, our model outperfom it in inference speed.
>
> | Models   | FPS (image/s) | Speed  |
> | -------- | -------------- | ---------------- |
> | CGFormer | 2.78           | 0.34x            |
> | PSALM    | 2.19           | 0.27x            |
> | ReMamber | 5.64           | 0.69x            |
> | **Ours** | **8.26**       | **1x**               |
>
> >**jD3V-Q2 && jD3V-W3:** What are the failure cases of the proposed method?: The paper could benefit from more analysis on failure cases of the proposed method. This would help clarify its limitations, especially in handling complex descriptions
>
> **jD3V-Q2-A && jD3V-W3-A:** Thank you for your question. We will add detailed visualization of failure cases in revised version. Here, we summarize below the main patterns observed in SaFiRe’s failure cases:
> - **Fine-grained attribute ambiguity.** Failures often occur when multiple candidates share broadly similar attributes but only one matches the intended semantics of the description. In such cases, even subtle differences in attributes (e.g., hand positioning) can carry decisive referential meaning, yet the model may overlook them due to visual similarity.
> *Example: “He crosses his hands while talking to the other person.”* In the image, one man rests one hand over the other, while another interlaces his fingers. Although both are hand-related gestures and might visually resemble “crossed hands,” only the latter accurately captures the intended meaning. The model, however, selects the former, revealing its difficulty in grounding fine-grained action attributes to textual descriptions.
>
> - **Long spatial/relational chain ambiguity.** Another common failure mode arises in texts involving long, compositional chains of spatial or relational references. These expressions require the model to perform multi-step understanding with frequent spatial shifts, jumping between different objects across the scene. A typical symptom is that the grounding process breaks down at one or more intermediate links, leading to cumulative error.
> *Example: “the mug on the tray beside the plate nearest the chef.”* In the image, there are multiple mugs, trays, and plates spread across a crowded table, and the chef is positioned off to one side. There is no obvious shortcut to the target—only by following the full relational chain can the correct object be identified. The model, however, misidentifies an object linked to an incorrect plate early in the chain, revealing its difficulty in maintaining spatial consistency and grounding excessive relational dependencies to textual descriptions.
>
> We will add the above analysis into revision to clarify the potential future directions.
>
> >**jD3V-W1:** The Saccade-Fixation design is novel; the rest of the model relies heavily on standard components (Swin-B, BERT). The innovation is primarily in the fusion mechanism rather than in the overall pipeline, which may limit its extensibility to other tasks or modalities.
>
>
> **jD3V-W1-A:**
> First, we sincerely thank you for recognizing the novelty of our Saccade–Fixation design. We would like to clarify that our choice of Swin-B and BERT as the backbone is primarily to ensure a fair comparison with mainstream methods. Meanwhile, it is a convention to focus on fusion mechanism designing in this field, as many other methods share the same backbones in Table 1 in our paper.
>
> Second, the Saccade–Fixation mechanism is a general-purpose fusion method; however, this work focuses on the RIS task, and we leave broader exploration to future work. For example, SaFiRe have the potential to be adapted to other task or modalities like Open Vocabulary Segmentation (OVS) by changing the task head with minimal changes, or Audio Visual Segmentation (AVS) by changing the backbones. We will mention such potential way of exploration in the conclusion as future work.
>
> >**jD3V-W2:** Although the introduced benchmark aRefCOCO increases ambiguity, it is still generated by a VLM, which could inadvertently introduce biases, even with meticulous filtering. The paper lacks evaluations on real-world, human-annotated ambiguous datasets or interactive grounding scenarios, limiting its external validity.
>
> **jD3V-W2-A:** We acknowledge this bias concern and have already claimed it in our limitations section (Appendix B). Since this study mainly focuses on the techinical innovation in designing a model to handle the referential ambiguity, we mainly verify our method in the common benchmark and our initial small set, aRefCOCO. For broader evaluations on real-world, and human annotated datasets as well as interactive grounding scenarios, we will conduct the continuous verification and extension in our future explorations.

---

> > ### Comment · Reviewer_jD3V · 2025-08-06
> >
> > I appreciate the authors’ detailed response. The rebuttal addresses most of my concerns. I will maintain my initial score.

---

> > > ### Author Response · Authors · 2025-08-06
> > > **Thank you for the response**
> > >
> > > Dear Reviewer jD3V,
> > >
> > > We sincerely appreciate your efforts in reviewing our submission and your thoughtful comments. Your suggestions are highly valued, and we will incorporate them into the final version of the paper. Thank you again for your valuable feedback and for helping us to improve our work.
> > >
> > > Best regards,
> > >
> > > Authors of Paper 2533

---

### Official Review · Reviewer_8mtA · 2025-07-02

**Clarity:** 2
**Significance:** 2
**Originality:** 3
**Rating:** 4
**Confidence:** 5

**Summary:**

This paper proposes SaFiRe, a novel framework for Referring Image Segmentation (RIS) that addresses referential ambiguity in natural language expressions. Inspired by human two-phase cognitive processes, SaFiRe integrates Saccade and Fixation operations using Mamba’s state-space modeling. The Saccade operation establishes coarse cross-modal alignment, while Fixation refines it via region-wise text reiteration. The authors also introduce aRefCOCO, a benchmark for ambiguous expressions, and demonstrate SaFiRe’s superiority over SOTA methods on both standard and aRefCOCO datasets. Key contributions include modeling cognitive mechanisms, efficient multi-cycle refinement with linear complexity, and a new benchmark for ambiguous RIS.

**Questions:**

Q1: Could the authors explicitly state the primary scientific concern that SaFiRe aims to address—beyond merely introducing a new framework? Given the experimental limitations (marginal gains over Magnet, and absence of recent SOTA benchmarks like EEVG/C3VG), it is unclear whether the core contribution lies in cognitive modeling or practical performance advancement.

Q2:Could the authors clarify whether the first SFLayer uses the final-layer features from Swin Transformer and BERT, and whether you use LAVT’s cross-modal fusion in the backbone stage?

Note: A thorough response that resolves these concerns and demonstrates the solidity of experimental validations could potentially influence the evaluation score positively.

**Ethical Concerns:**

["NO or VERY MINOR ethics concerns only"]

**Final Justification:**

I raise the score in this stage.

**Limitations:**

yes

**Paper Formatting Concerns:**

Not Involved

**Quality:**

3

**Strengths And Weaknesses:**

Strengths:
1.The paper introduces SaFiRe, a novel framework that simulates human two-phase cognitive processes: global understanding via Saccade and detailed refinement via Fixation, to address referential ambiguity in RIS. By leveraging Mamba’s state-space modeling, the framework achieves efficient multi-cycle refinement with linear complexity, enabling robust alignment between visual regions and ambiguous language expressions. This design achieves significant performance gains on standard benchmarks and superior accuracy on ambiguous expressions.

2.The authors present aRefCOCO, a test-only benchmark specifically designed to evaluate RIS models under object-distracting and category-implicit scenarios. With an average of 12.6 words per expression, 3.1 same-category distractors per image, and frequent pronoun usage, aRefCOCO pushes the field to tackle real-world referential ambiguity.

Weaknesses:
1.Unfair, Marginal and Incomplete Comparative Evaluation
The experimental validation lacks rigor due to suboptimal benchmarking choices. SaFiRe uses Swin Transformer as the visual backbone rather than Mamba, making direct comparisons with ReMamber (a pure Mamba-based model) inherently unfair, as the latter’s architecture is tailored for state-space modeling . When compared to Magnet (a Swin-BERT-based method more aligned with SaFiRe’s backbone), performance gains are marginal, failing to demonstrate substantial superiority. Additionally, the evaluation omits recent SOTA methods(e.g., EEVG[1], C3VG[2], OneRef[3], and PromptRIS[4]). This absence hinders the paper’s ability to validate SaFiRe’s competitiveness in the evolving RIS landscape, casting doubt on the robustness of its claimed advancements.

2.Inadequate Transparency in aRefCOCO Dataset Generation
The reliability of aRefCOCO, the proposed benchmark for ambiguous expressions, is undermined by insufficient evidence of its annotation process. While the authors mention that a significant portion of text-image pairs were generated using Qwen-VL-Max, they omit critical details such as the prompting strategies, filtering criteria, and validation mechanisms . The absence of concrete examples (e.g., sample prompts, annotation guidelines) hinders reproducibility and raises concerns about potential biases introduced by LLM-generated texts. Without transparent documentation, it is difficult to assess whether the dataset truly captures real-world referential ambiguity or inadvertently reflects the limitations of the underlying language model .

[1] An efficient and effective transformer decoder-based framework for multi-task visual grounding. In ECCV, 2024.
[2]Multi-task visual grounding with coarse-tofine consistency constraints. In AAAI, 2025.
[3]Oneref: Unified one-tower expression grounding and segmentation with mask referring modeling. In NeurIPS, 2024.
[4]Prompt-driven referring image segmentation with instance contrasting. In CVPR,2024.

---

> ### Author Rebuttal · Authors · 2025-07-31
>
> Thank you for the detailed review. We would like to first thank you for highlighting the novelty and performance gains of our Saccade–Fixation framework—benefits that R7DT and jD3V likewise praised, with even Wg1a calling the experimental results “promising.”
>
> Below, we address each of your comments point‑by‑point and clarify how our revisions and additional experiments resolve the concerns you raised.
>
> >**8mtA-W1:** Concerns regarding comparative evaluation: (1) "Unfair" comparison with ReMamber due to backbones difference; (2) "Marginal" gains over Magnet; (3) Missing recent SOTAs
> ### (1) Comparing with ReMamber
> We include ReMamber not for a direct comparison of performance, but rather **as a reference baseline**, since it is the only published Mamba-based RIS method. We will revise our tables to clearly indicate that ReMamber is a full-Mamba-based architecture, to avoid any misunderstanding regarding the fairness of the comparison. Meanwhile, as other methods, ReMamber still approaches RIS through keyword-matching modeling and cannot effectively address the referential ambiguity problem that we aims to solve.
> ### (2) Comparing with MagNet
> SaFiRe demonstrates consistent gains across all datasets of RefCOCO series. Our advantage on RefCOCO is kind of marginal because this dataset is relatively simple. However, as the dataset becomes more challenging, our model's advantage becomes evident, with strong improvements on the most difficult **RefCOCOg** dataset (+1.3 oIoU on average).
>
> Additionally, we here provide more comparisons with MagNet, including mixed‑dataset training (which better reflects the linguistic variety than single-dataset) and inference speed. Results are averaged over each split, with the numbers in parentheses indicating the **performance gap relative to ours**. SaFiRe surpasses MagNet in both accuracy and speed.
>
> |Method|Backbone|Metric|RefCOCO|RefCOCO+|RefCOCOg|
> |-|-|-|-|-|-|
> |MagNet|SwinB+BERT|oIoU|75.7(-5.4)|67.9(-6.1)|68.5(-7.8)|
> |**SaFiRe (Ours)**|SwinB+BERT|**oIoU**|**81.1**|**74.0**|**76.3**|
>
> |Models|FPS (images/s)|Speedup|
> |-|-|-|
> |MagNet|3.55|1x|
> |**Ours**|**8.26**|**2.32x**|
>
> ### (3) Comparing with recent SOTAs
> First, we want to clarify that the main evaluation metric reported in [1]-[4] is **mIoU**, which differs from the widely-used **oIoU** in this field. mIoU is typically numerically ***higher*** than oIoU (see results in [2],[4]). The distinction between the two is shown in LAVT(CVPR2022).
> Here, we provide the comparison using mIoU and oIoU. Note that their main results are under mixed-dataset setting (similar with Tab.2 in our main paper). Results are averaged over each split, with the numbers in parentheses indicating the **performance gap relative to ours**. Compared with these methods, SaFiRe still achieves the best accuracy.
>
> |Method|Backbone|Metric|RefCOCO|RefCOCO+|RefCOCOg|
> |-|-|-|-|-|-|
> |EEVG [1]|Swin-B+BERT|mIoU|77.5(-4.5)|70.5(-6.4)|71.7(-6.0)|
> |PromptRIS [4]|CLIP+SAM|mIoU|78.0(-4.0)|70.7(-6.2)|69.8(-7.9)|
> |OneRef-B (ft)[3]|BEiT3-B|mIoU|79.6(-2.4)|74.1(-2.8)|74.5(-3.2)|
> |C3VG [2]|BEiT3-B|mIoU|81.1(-0.9)|76.3(-0.6)|76.7(-1.0)|
> |**SaFiRe (Ours)**|Swin-B+BERT|**mIoU**|**82.0**|**76.9**|**77.7**|
>
> |Method|Backbone|Metric|RefCOCO|RefCOCO+|RefCOCOg|
> |-|-|-|-|-|-|
> |PromptRIS [4]|CLIP+SAM|oIoU|76.3(-4.8)|66.5(-7.5)|66.0(-10.3)|
> |C3VG [2]|BEiT3-B|oIoU|80.6(-0.5)|73.8(-0.2)|75.4(-0.9)|
> |**SaFiRe (Ours)**|Swin-B+BERT|**oIoU**|**81.1**|**74.0**|**76.3**|
>
> (ft denotes a further fine-tuning is performed on the specific dataset.)
>
> Second, since C3VG and EEVG provide codes and weights, we further test them on aRefCOCO; SaFiRe keeps a clear lead on all metrics.
>
> |Method|oIoU|mIoU|Pr@50|Pr@70|Pr@90|
> |-|-|-|-|-|-|
> |EEVG [1]|50.7|59.4|66.9|62.4|29.4|
> |C3VG [2]|62.9|69.1|78.4|72.8|32.8|
> |**SaFiRe (Ours)**|**65.1**|**71.4**|**78.9**|**73.6**|**45.2**|
>
> Third, given C3VG’s good accuracy, we also compare inference speed: SaFiRe runs 3.3× faster, confirming its advantage in application.
>
> |Models|FPS (images/s)|Speedup|
> |-|-|-|
> |C3VG [2]|2.53|1x|
> |**Ours**|**8.26**|**3.27x**|
>
> >**8mtA-W2:** Inadequate Transparency in aRefCOCO Dataset Generation. The reliability of aRefCOCO, including VLM prompts,  filtering and validation steps.
> - **8mtA-W2-A:** Here we provide details of this process. Codes and datasets will be released upon publication.
> ### (1) Annotation Prompting
> In order to ensure the richness and diversity of object descriptions while adhering to the fundamental requirement of single-object localization in RIS tasks, we used a prompt template for Qwen-VL-Max as below:
>
>     "### Role\n"
>     "You are a professional assistant, proficient in object recognition and description. Your task is to provide a clear and focused description of the main object inside the red box in the image. You are given the category information of the main object inside the red box as '{target}'. The description should be based on the provided target information, and you should generate a more detailed description with your visual analysis of the image. Please note that there is no visible box in the actual image.\n\n"
>
>     "### Target Category Information\n"
>     f"targt category information: '{target}'\n\n"
>
>     "### Skills\n"
>     "1. [Important!] Please only focus on the main object, i.e., '{target}', and ignore the background or irrelevant details.\n"
>     "2. [Important!] Do not assume or include any information that is not present in the image content.\n"
>     "3. [Important!] Provide 5 independent English sentences, each of which can independently identify '{target}' without relying on other sentences. Each sentence should describe a unique identifying feature.\n"
>     "4. [Important!] Use the subject '{target}' from the targt category information, and ensure you are describing a single object '{target}'! Others should be able to locate only one object based on your sentences.\n"
>     "5. [Important!] Use singular language to ensure the description focuses only on one main object. Do not use plural words like 'they' as the subject, but you are allowed to use pronouns as the subject.\n"
>     "6. [Important!] If you need to describe based on objects around '{target}', please keep the subject as '{target}'.\n\n"
>
>
> ### (2) Filtering
> We first pre-screen all descriptions with GPT 4o for the two patterns:"object-distracting" (i.e., containing multiple nouns) and "category-implicit" (i.e., where the subject is a pronoun). Items that match either pattern are retained.
> Then, to verify the remaining pairs meet the basic rules we set for annotation, GPT 4o is involved as an evaluator. The prompt is as follows:
>
>     Role:
>     You are a professional visual-language evaluator. Your task is to verify whether each given description sentence is a faithful, grounded, and rule-abiding description of the target object **inside the red box** in the provided image.
>
>     Information:
>     The target object belongs to the category: {target}.
>
>     The description was generated under the following constraints:
>     1. It must describe only the main object ('{target}'), ignoring the background or other irrelevant elements.
>     2. It must not include objects that are not in the whole image, but objects out of box are allowed.
>     3. Each sentence must independently identify the target.
>     4. It can contains objects beyond the target.
>     5. It must use singular phrasing, and refer to the object using either its category label ('{target}') or singular pronouns.
>
>     You will be given 5 sentences. Your task is to verify, for each sentence, whether it satisfies **all** the above requirements based solely on the image content.
>
>     Sentences to Verify:
>     {Sentences}
>
>     Instructions:
>     1. For each sentence, respond with **"yes"** or **"no"**. if "no", follow it by a **brief reason**.
>     2. If any rule is violated or the content is ungrounded, answer "no" and specify which rule(s) are broken.
>     3. Do not use external knowledge or assumptions. Base your judgment strictly on what is visible in the image.
>     4. Provide one answer per sentence, in order. Each answer must be on its own line.
>     5. Do not add commentary outside the answers. Return only the answers.
>     6. Ensure the number of answers equals 5.
>
> ### (3) Validation
> Manual review was conducted by at least two annotators with RIS domain expertise. Each pair was independently validated to confirm (1) whether the description uniquely refers to a single object in the image, and (2) whether the expression obeys our referential ambiguity criteria while remaining image-groundable. In borderline cases, annotators reached consensus through discussion, and a third reviewer was consulted when needed.
>
> > **8mtA-Q1:** Given the experimental limitations, state the primary scientific concern that SaFiRe aims to address
>
> **8mtA-Q1-A:** First, we hope the experiments and analysis in the above response (to 8mtA-W1) can **resolve the reviewer's concern about experimental limitations**. Our method surpass all existing RIS models, demonstrating its superiority.
>
> Second, we want to clarify that our goal is to **resolve referential ambiguity**, a fundamental challenge in real-world RIS scenarios. Previous methods struggle in such scenarios due to their **reliance on keyword-matching modeling**. Instead, we **jump out of such paradigm** and propose our Saccade-Fixation mechanism to address the issue. To further demonstrate this point, a more challenging benchmark aRefCOCO is proposed besides traditional benchmarks.
>
> >**8mtA-Q2:** clarify whether the first SFLayer uses the final-layer features from Swin and BERT, and whether use LAVT’s cross-modal fusion in the backbone stage
>
> **8mtA-Q2-A:** Yes, the first SFLayer uses only the final-layer features from both the Swin Transformer and BERT. We also adopt LAVT’s cross-modal fusion in the backbone stage as CGFormer (CVPR23), SADLR (AAAI23), and ReLA (CVPR23) do. We will clarify these details in the revision.

---

> > ### Comment · Reviewer_8mtA · 2025-08-05
> > **After Rebuttal**
> >
> > Thanks for the rebuttal, which solved most of my concerns. I will raise the score. I hope the authors can compare with more recent SOTA methods mentioned in the rebuttal in the final revised version.

---

> > > ### Author Response · Authors · 2025-08-06
> > > **Thank you for the response**
> > >
> > > Dear Reviewer 8mtA,
> > >
> > > We sincerely appreciate your constructive review and the valuable feedback, which helps further improve our submission. We will carefully follow your advice and incorporate the discussion and experiments including comparisons with more recent SOTA methods into the revision.
> > >
> > > Best regards,
> > >
> > > Authors of Paper 2533

---

### Decision · Program_Chairs · 2025-09-17

**Decision:**

Accept (poster)

**Comment:**

The paper introduces SaFiRe, a framework for Referring Image Segmentation (RIS) inspired by human cognitive processes, combining a global “Saccade” stage with a fine-grained “Fixation” stage, supported by Mamba’s scan-update mechanism. It also proposes aRefCOCO, a new benchmark designed to test RIS under ambiguous expressions, including object-distracting and category-implicit scenarios. Experiments on standard datasets and aRefCOCO demonstrate clear improvements over existing methods.

According to reviewers, strengths include the novel cognitive-inspired design, effective use of Mamba for efficient iterative refinement, and the valuable benchmark contribution which targets a genuine gap in RIS evaluation. The paper is generally well written, with ablation studies supporting its claims.

However, several weaknesses remain. Comparisons are incomplete: recent SOTA methods (e.g., EEVG, C3VG, LISA, PromptRIS) are missing, which makes competitiveness less clear. The novelty of SaFiRe’s mechanism is debated, with some components seen as adaptations of existing designs. The generation of aRefCOCO relies partly on VLM outputs without sufficient transparency, raising concerns about reproducibility and dataset validity. Also, the gains over Magnet are sometimes marginal, and distinctions from related reasoning segmentation frameworks (e.g., LISA) are not fully articulated.

During rebuttal, the authors clarified aspects of model design and dataset construction, and addressed fairness of comparisons, though concerns about broader benchmarking and dataset documentation were only partially resolved. On balance, I think the methodological idea and benchmark are timely and solid despite limitations. The contributions warrant acceptance, but revisions should strengthen all issues aforementioned.